# A Case Study of a 10-Year Change in the Vegetation and Water Environments of Volcanic Mires in South-Western Japan

**Akira Haraguchi**

Faculty of Environmental Engineering, The University of Kitakyushu, Hibikino 1-1, Wakamatsu, Kitakyushu 808 0135, Fukuoka, Japan; akhgc@kitakyu-u.ac.jp

**Abstract:** Variations in the groundwater environments and dominant species of volcanic mire vegetation were monitored for 10 years in a volcanic area in south-western Japan. The correlation between changes in groundwater environments and vegetation revealed that changes in water environments determine the dominant species of volcanic mire vegetation. The amount of spring water supplied to the mire vegetation determines the water-table depth and the subsequent nutrient supply. The *Sphagnum* spp. coverage decreased with increasing base cation concentrations, particularly the $Ca^{2+}$ concentration up to 40 mg/L. The *Moliniopsis japonica* coverage increased with the decreasing *Sphagnum* spp. coverage. The nutritional variables of water supplied to vegetation affected by volcanic activity changed the type of dominant species. A 10-year change in vegetation in the volcanic mires revealed that vegetation succession in volcanic mires evolved from ombrogenous to minerogenous and from minerogenous to ombrogenous communities. The water environment promoted changes in the dominant species.

**Keywords:** *Sphagnum*; succession; volcano; peatland; water chemistry

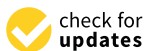



## 1. Introduction

The vegetation of volcanic mires has been studied with reference to the surface soil and groundwater hydrological and nutritional conditions. Water chemistry and water-table depth are the primary variables determining species composition in volcanic mires [1,2]. The nutrients that the bedrock supplies to the peat surface decrease with increasing peat depth, and at the onset of the mire, the vegetation changes from minerotrophic to ombrotrophic based on the succession progression from fens to bogs [3]. *Sphagnum* species, dominant in ombrotrophic mires, typically dominate bog communities [4] and are the most prominent boreal mires.

Natural and artificial disturbances cause changes in mire vegetation. Tahvanainen [5] found ombrotrophication in a boreal aapa mire owing to hydrological disturbances in the catchment. Mire drainage is a significant artificial disturbance to mire vegetation, and some studies have documented the vegetation recovery process after restoration. For example, Haapalehto et al. [6] revealed the successful vegetation recovery of drained peatlands after approximately 10 years of rising water table by blocking drainage ditches. The mire vegetation recovery in extracted peatlands is an important subject in mire restoration. Kozlov et al. [7] monitored the revegetation process after 15 years of rewetting extracted peatlands. Studies on vegetation recovery and the revegetation of mires suggest that vegetation monitoring is required for at least 10 years to determine changes in vegetation in disturbed mires.

Landslides cause paludification through the accumulation of sediments blocking the stream water flow, thus, initiating the formation of mires in the mountainous region [8]. In volcanic mountainous regions, volcanic activity naturally disturbs mire vegetation distribution. Changes in vegetation owing to the direct and indirect effects of volcanic ash deposition have been reported. Thick layers of volcanic ash deposition

can damage vegetation. The deposited volcanic ash forms a less permeable horizon, changing the hydrological conditions of the mire [9], and causes chemical changes in the topsoil. Wolejko and Ito [10] proposed the tephra trophic concept, which explains a change in vegetation caused by the deposited tephra mineralization. Hotes et al. [11] reported that the volcanic ash layer could improve soil aeration and the subsequent mineralization of nutrients after the decomposition of organic materials in the surface soil. A previous study on a volcanic mire in south-western Japan on the chronological analysis of macrofossils in the peat layer revealed a repeated change in vegetation between minerotrophic and ombrotrophic vegetation after the deposition of the most recent volcanic ash (970 ± 40 y.B.P) [12]. Information on the effects of tephra deposition on the nutritional status of mire vegetation is lacking; however, volcanic ash deposition, volcanic gas, landslide, and groundwater are known to be affected by volcanic activity in mires in the volcanic region catchment area.

In the present study, vegetation changes in two mires in the Kujyu volcanic area in south-western Japan were investigated, considering groundwater depth and surface water chemistry. The present study aimed to determine variations in mire vegetation and the water environment over 10 years and reveal the correlation between changes in vegetation and the water environment variables.

## 2. Materials and Methods

### 2.1. Study Sites

The Tadewara mire (38 ha, 1017 m a.s.l.) is in the north facing a gentle slope under the knick points of Mt. Mimata, Mt. Hossho, and Mt. Sensui in Oita Prefecture, south-western Japan (Figure 1). *Miscanthus sinensis* Anders is dominant on the upper parts of the slope (southern part), and *Phragmites australis* Cav. (Trin.) ex Steud.; *Moliniopsis japonica* (Hack.) Hayata.; and *Hydrangea paniculata* Sieb. et Zucc. are dominant on the lower part of the slope (northern part). *Sphagnum palustre* L. and *Sphagnum fimbriatum* Wils. communities are established on the ground surface underneath the *P. australis* and *M. japonica* canopies. Domes typical for bogs develop at the lower end of the slope, and hummocks 30–50 cm high are distributed on the dome. Streams originate from springs flowing at the eastern and western sides of the mire.

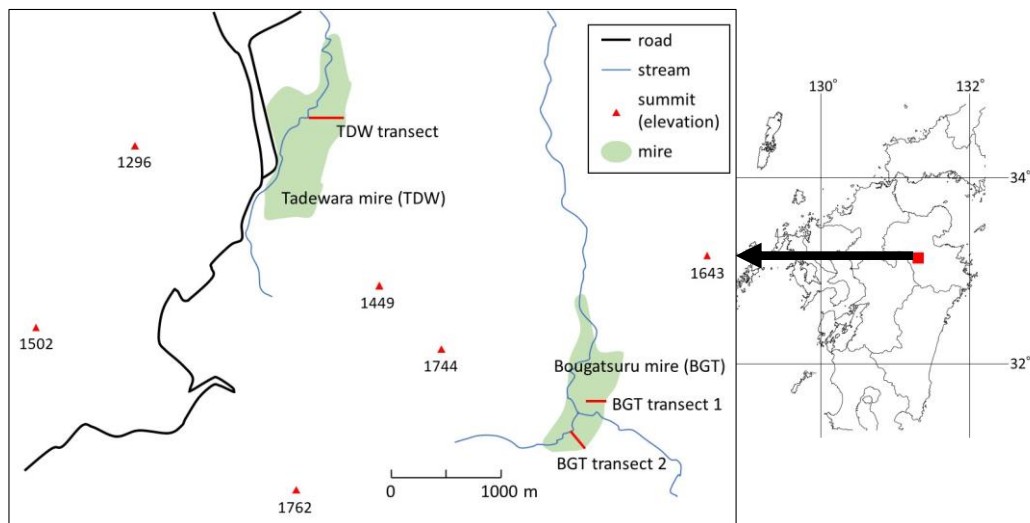

**Figure 1.** Map showing the Tadewara mire (TDW) and Bougatsuru mire (BGT) in south-western Japan. TDW transect, BGT transect 1, and BGT transect 2 are investigated transects with a length of 160 m, 70 m, and 90 m, respectively.

The Bougatsuru mire (53 ha, 1240 m a.s.l.) is located at the ground level under the knick points of Mt. Mimata and Mt. Taisenzan in Oita Prefecture, south-western Japan (Figure 1).

The Bougatsuru mire is divided in two by a stream (the origin of the Chikugogawa River). The northern part of the Bougatsuru mire spreads west facing a gentle slope with scattered bedrocks on the ground surface. Seepage water is directed to the lower part of the slope. The upper part of the slope is dominated by *M. sinensis*. Contrastingly, the lower part is dominated by *M. japonica*, *S. palustre*, and *S. fimbriatum*. The southern part of the Bougatsuru mire spreads west facing a gentle slope, with springs at the upper end of the slope. The mire vegetation is dominated by *M. japonica*, *S. palustre*, *S. fimbriatum*, and *P. australis*.

## 2.2. Groundwater Table and Water Chemistry

A 160 m transect line with 17 sampling sites from 33.12310 N; 131.23523 E; and 1014 m a.s.l (0 m site) to 33.12267 N; 131.23689 E; and 1019 m a.s.l (160 m site) was established in the Tadewara mire (TDW transect; Figure 1). The TDW transect was across the center of the dome and dominated by *Sphagnum* spp. Two transect lines were established in the Northern and Southern Bougatsuru mire. Bougatsuru transect 1 in the northern part of the Bougatsuru mire (BGT transect 1; 70 m long) from 33.10003 N; 131.26279 E; and 1238 m a.s.l (0 m site) to 33.10086 N; 131.26188 E; and 1231 m a.s.l (70 m site) was established. The Bougatsuru transect line 2 in the southern part of the Bougatsuru mire (BGT transect 2; 90 m long) was from 33.09637 °N; 131.26065 °E; and 1248 m a.s.l (0 m site) to 33.09693 °N; 131.26012 °E; and 1241 m a.s.l (90 m site).

Piezometers comprising vinyl chloride pipes (1.3 m long and 38 mm in diameter) with 2 mm holes at 10 cm intervals, on both sides, were inserted vertically to the ground surface at approximately 90 cm depth, at 10 m intervals, on every transect in June 2006. The distance from the ground surface to the water table was measured monthly at each site and every transect. Water samples were collected from each pipe on the same day as the groundwater level measurement. The electrical conductivity (EC) and pH were measured in the laboratory using EC and pH meters equipped with EC and pH probes (D-54 and D-74, Horiba, Kyoto, Japan). The total organic carbon (TOC) for a non-filtered water sample was determined using a TOC analyzer (TOC-VCSH, Shimadz, Kyoto, Japan). Total nitrogen (TN) was measured using UV absorption spectrophotometry after digestion with potassium persulfate. Total phosphorus (TP) was measured using the molybdenum blue method after digestion with potassium persulfate. The samples were filtered through 0.20 μm cellulose acetate membrane filters (DISMIC-25, Advantec Toyo, Tokyo, Japan). The major cations ($NH_4^+$, $Na^+$, $K^+$, $Mg^{2+}$, and $Ca^{2+}$) and anions ($Cl^-$, $NO_2^-$, $NO_3^-$, $PO_4^{3-}$, and $SO_4^{2-}$) were analyzed using an ion chromatograph (DX-120 and ICS-2100, Thermo Scientific Dionex, Thermo Fisher Scientific, Tokyo, Japan).

Groundwater table measurements and water sampling were performed from June 2006 to May 2017. ata from August 2008 to May 2010 were not presented in the TDW transect, and data from August 2007 to July 2010 were not presented in BGT transects 1 and 2. Additionally, data recorded under storm and heavy snow conditions were not presented.

## 2.3. Vegetation

Along the TDW transect, BGT transect 1, and BGT transect 2, 1 × 1 m quadrats were sequentially placed at 160, 70, and 90 sites, respectively. The coverage of each plant species in each quadrat was recorded in 2006, 2007, 2008, 2010, 2011, 2013, and 2016 in the TDW transect and in 2006, 2010, 2011, 2013, and 2016 in BGT transects 1 and 2. Vegetation coverage was determined in July and August.

## 2.4. Topography

The relative elevation on each transect line was measured using a digital theodolite (DT-114, Topcon, Tokyo, Japan) at 1 m intervals in the TDW transect in 2008, 2010, and 2017 and in BGT transects 1 and 2 in 2010 and 2017.

*2.5. Data Analysis*

Data on the water environment and vegetation were divided into two periods: the first five years from 2006 to 2011 and the last five years from 2011 to 2016, and these were analyzed.

The changes in water environment variables were tested using linear regression. Chemical variables and ground surface data from 2006 to 2011 and from 2011 to 2016 were separately used for a regression analysis for the TDW transect. Contrastingly, only data from 2011 to 2016 were used for BGT transects 1 and 2 because of the prolonged period of missing data from 2006 to 2011 in these transects. The linear regression of each variable (dependent variables) against elapsed days from the beginning of data collection (independent variables) was analyzed. The slope of the linear regression was tested using an analysis of variance.

Data from 2006, 2011, and 2016 were used to analyze vegetation change, and the differences in the coverage of each abundant species were tested. Quadrats within 5 m of a data collection site of the water environment (position of every piezometer) were assigned as representatives of that measurement site; thus, there were five quadrats for each measurement site at both ends of each transect and 10 quadrats for the non-terminal measurement sites of each transect. The coverage of each abundant species in each representative quadrat was used for analysis. Species with a coverage of less than 30% were excluded. Significant differences in the coverage of each species at each measurement site between 2006 and 2011 and between 2011 and 2016 were tested using a Wilcoxon test.

The correlation between the water environment and vegetation changes was tested. A significant increase in the water environment and vegetation from 2006 to 2011 and from 2011 to 2016 was assigned a value of 1, whereas a significant decrease in these variables was assigned a value of −1. Variables without significant differences were considered missing values. The significance of the correlation was tested using Spearman's test.

**3. Results**

Data on the water environment and vegetation changes at the TDW transect and BGT transects 1 and 2 are presented in https://doi.org/10.13140/RG.2.2.34825.31848.

*3.1. Topography*

The center of the dome of the TDW transect was approximately 1.0 m higher than those of both ends of the TDW transect (Figure 2). Streams at the western end of the TDW transect (0 m site) constantly flowed throughout the study period, whereas that in the eastern end of the transect line (site 17) was filled with sand after the landslide in 2007. The TDW transect was approximately at the center of the dome, and the roughness of the peat surface revealed well-developed hummocks. The slope of BGT transect 1 was approximately 5.0%. The eastern end (0 m site) of BGT transect 1 was approximately 3.5 m higher than that of the western end (70 m site (Figure 2)). The slope of BGT transect 2 was approximately 4.4%. The south-eastern end (0 m site) of BGT transect 2 was approximately 4.0 m higher than that of the north-western end (90 m site). BGT transects 1 and 2s'surfaces were almost flat without discernible microtopography.

*3.2. Vegetation Change Overview*

The coverage of abundant species in each 1 × 1 m quadrat is shown as a contoured figure on the axes of year and position (Figures 3–5). Missing data were interpolated, and contour figures were drawn by R version 4.2.1 [13].

The vegetation on the TDW transect in 2006 was dominated by two *Sphagnum* spp. and *M. japonica* (Figure 3). The distribution ranges of the two *Sphagnum* spp. on the TDW transect were distinct. *P. australis* and *H. paniculata* were the dominant species in some parts of the transect. The *Sphagnum* spp. coverage decreased from 2006 to 2010 and then increased until 2011. The *M. japonica* coverage was almost constant from 2006 to 2009 and then decreased until 2012, corresponding to an increase in *Sphagnum* spp. coverage. The *P.*

*australis* and *H. paniculata* coverage decreased from 2006 to 2012, except at the 140–160 m sites, and increased from 2012 to 2016. *M. japonica* coverage increased and that of *Sphagnum* spp. decreased from 2012 to 2016; therefore, the *Sphagnum* spp.-dominated vegetation changed to *M. japonica*-dominated vegetation from 2012 to 2016. The *Juncus decipiens* coverage increased from 2006 to 2016, particularly after 2013.

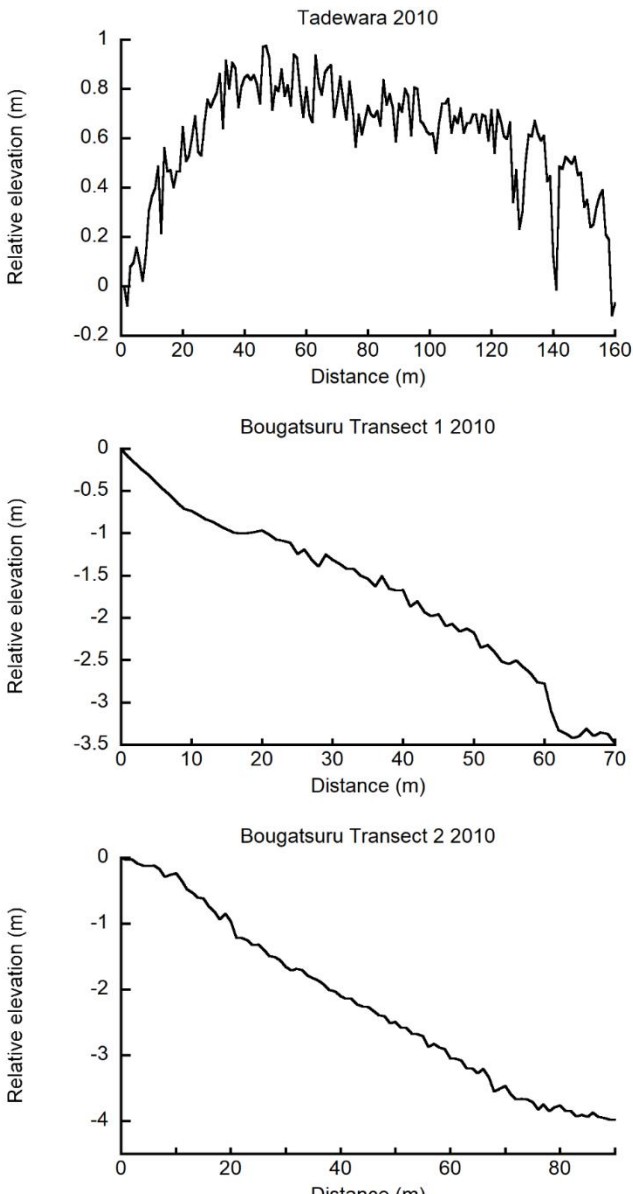

**Figure 2.** Elevation of ground surface relative to the origin (0 m site) of Tadewara transect (TDW transect), Bougatsuru transect 1 (BGT transect 1), and Bougatsuru transect 2 (BGT transect 2). Data were collected in July 2010.

*Sphagnum* spp. coverage temporally decreased in 2010–2012 (with some exceptions). Contrastingly, that of *M. japonica* increased during the same period at BGT transect 1 (Figure 4). BGT transect 1 vegetation changed temporarily from *Sphagnum*-dominated to *M. japonica*-dominated vegetation; however, the *S. fimbriatum* coverage increased until 2016. The *P. australis* coverage decreased from 2006 to 2012 and then increased from 2012 to 2016. The *J. decipiens* coverage increased from 2011 to 2016.

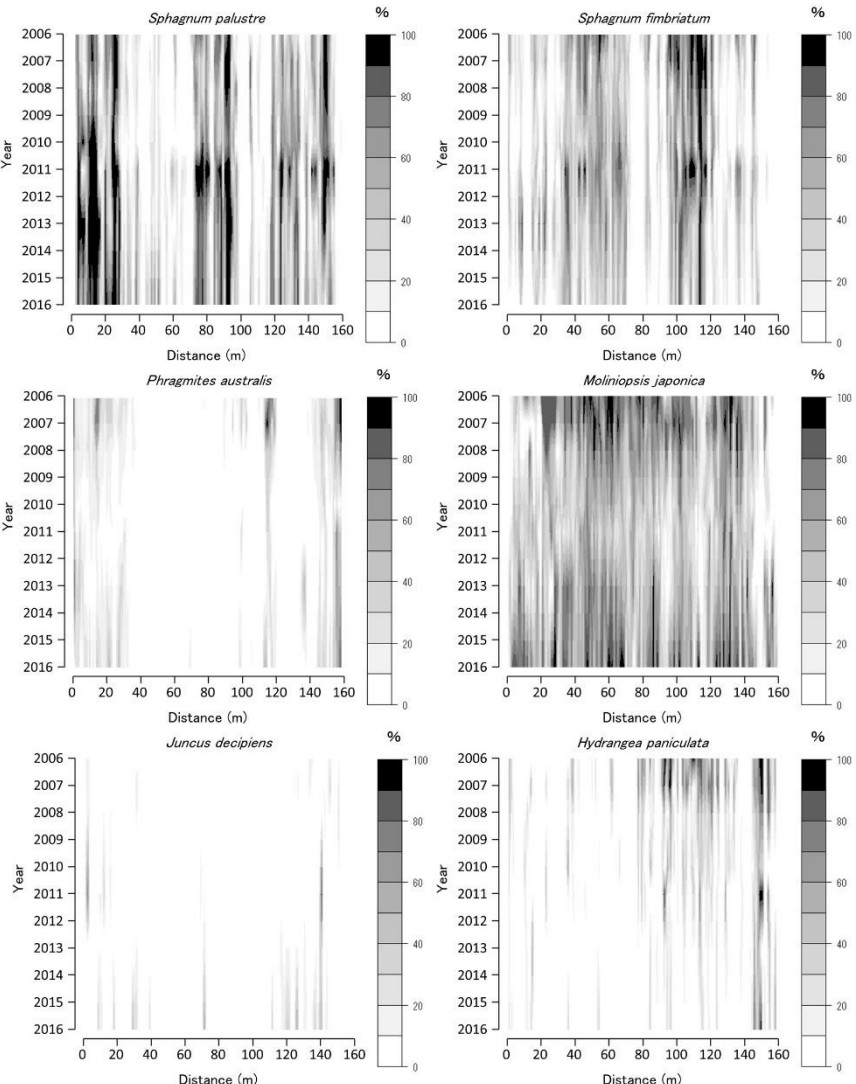

**Figure 3.** Coverage of *Sphagnum palustre*, *Sphagnum fimbriatum*, *Phragmites australis*, *Moliniopsis japonica*, *Juncus decipiens*, and *Hydrangea paniculata* along the Tadewara transect (TDW transect). Data were collected at every $1 \times 1$ m$^2$ quadrat placed sequentially on the transect. Countered figures of coverage on the axes of year and position were drawn by interpolating data collected in 2006, 2007, 2008, 2010, 2011, 2013, and 2016.

The coverage of the two *Sphagnum* spp. decreased from 2006 to 2011 and increased from 2012 to 2016, particularly at the upper part (10–20 m sites) of BGT transect 2 (Figure 5). The *P. australis* coverage increased, corresponding to a decrease in *Sphagnum* spp. coverage from 2011 to 2012. The *M. japonica* coverage decreased from 2006 to 2010 and then increased from 2012 to 2016. The *J. decipiens* coverage increased from 2012 to 2016 and was consistently high at the 80 m site of the transect line. The *P. thunbergii* coverage increased from 2006 to 2016, particularly at the upper part of BGT transect 2. Vegetation on BGT transect 2 changed from *Sphagnum*-dominated to *P. australis*-dominated vegetation on the upper part of the slope.

*3.3. Water Environment Change Overview*

The annual average values of the selected water environmental variables at each monitoring site are shown as contour figures on the axes of year and position (Figures 6–8). Missing data were interpolated, and contour figures were drawn by R version 4.2.1 [13]. Among the variables of the water environment, water-table depth (WTD), pH, EC, and $Ca^{2+}$, $SO_4^{2-}$, and TOC concentrations were selected and are shown in Figures 6–8. Among

these selected variables, pH and EC are general water chemistry variables, $Ca^{2+}$ inhibits *Sphagnum* growth [14], and $SO_4^{2-}$ is the primary anion in volcanic spring water [15].

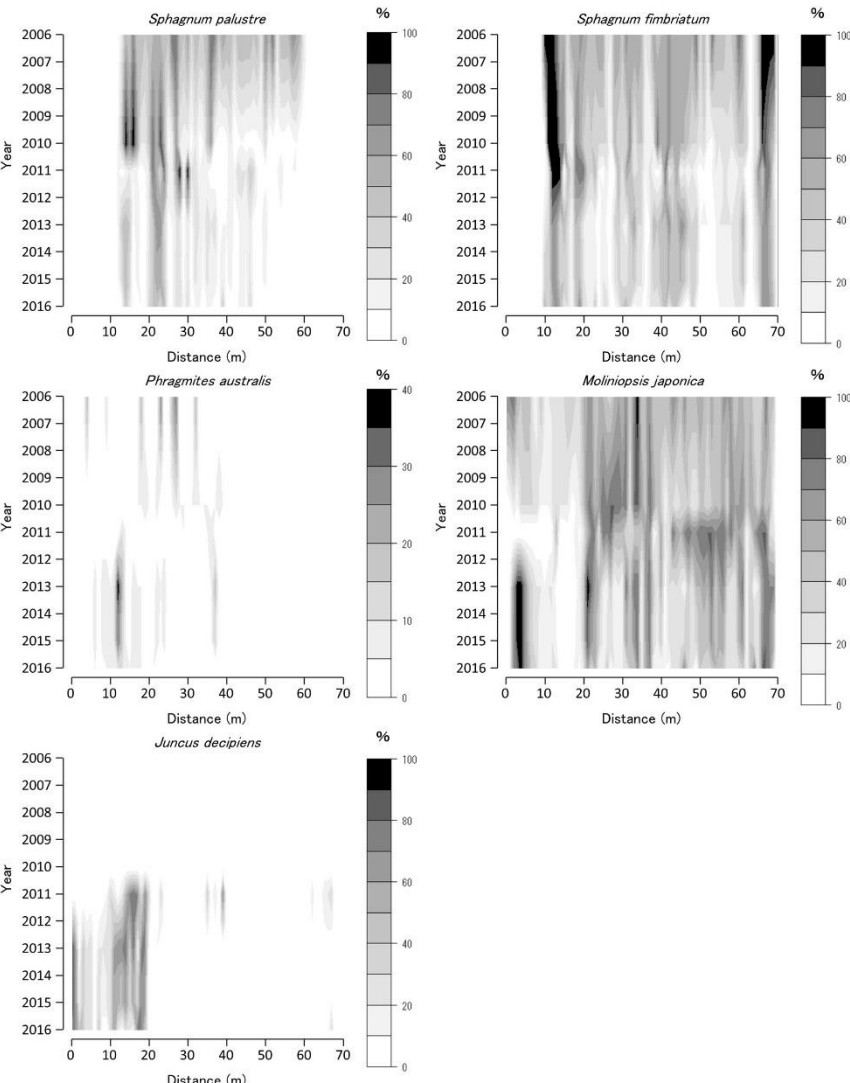

**Figure 4.** Coverage of *Sphagnum palustre*, *Sphagnum fimbriatum*, *Phragmites australis*, *Moliniopsis japonica*, and *Juncus decipiens* along Bougatsuru transect 1 (BGT transect 1). Data were collected at every $1 \times 1$ m$^2$ quadrat placed sequentially on the transect. Countered figures of coverage on the axes of year and position were drawn by interpolating data collected in 2006, 2010, 2011, 2013, and 2016.

The 0 m site of the TDW transect was located next to the stream and was constantly inundated from 2006 to 2016 (Figure 6). The WTD on the TDW transect at the 20, 60, and 100 m sites was higher than those at the other transect sites that were inundated in 2006. The 20 m site was constantly inundated from 2006 to 2013. WTD decreased from 2011 to 2014 and increased from 2014 to 2016 at all the TDW transect sites. Lower pH values were observed at the 40 and 80 m TDW transect sites than those at the other TDW transect sites. The pH increased from 2006 to 2008, decreased from 2008 to 2011, increased from 2011 to 2013, decreased from 2013 to 2014, and then increased from 2014 to 2016. The EC was higher at the TDW transect 0–20 m and 40–100 m sites. The EC values decreased from 2009 to 2016. $Ca^{2+}$ concentration was higher at the TDW transect 0–80 m sites than those of the other TDW transect sites and was the highest in 2008 and 2009. $Ca^{2+}$ concentration decreased from 2010 to 2016. The concentration of $SO_4^{2-}$ was higher at the TDW transect 0–100 m sites than those at other TDW transect sites and was the highest in 2010. TOC was

higher at the TDW transect 120–160 m sites than that at other TDW transect sites and was the highest in 2012.

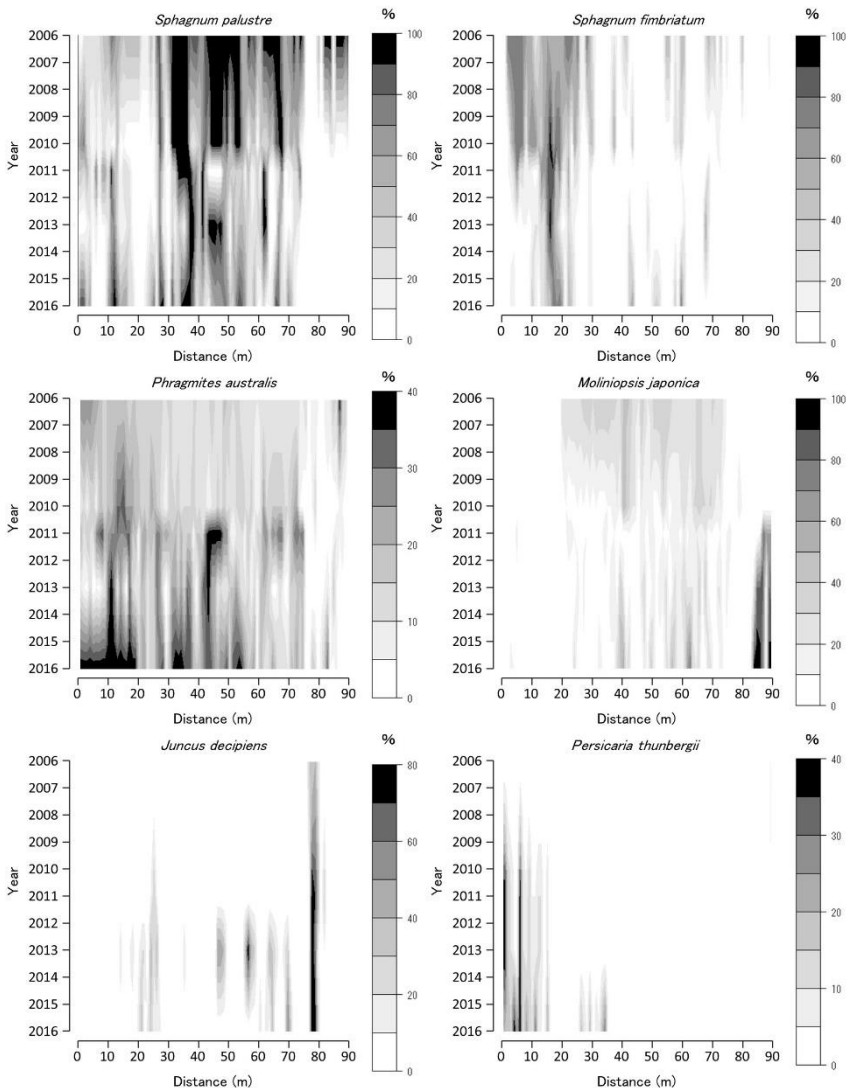

**Figure 5.** Coverage of *Sphagnum palustre*, *Sphagnum fimbriatum*, *Phragmites australis*, *Moliniopsis japonica*, *Juncus decipiens*, and *Persicalia thunbergii* along Bougatsuru transect 2 (BGT transect 2). Data were collected at every 1 × 1 m² quadrat placed sequentially on the transect. Countered figures of coverage on the axes of year and position were drawn by interpolating data collected in 2006, 2010, 2011, 2013, and 2016.

The WTD in BGT transect 1 exhibited a negative value consistently, implying that the water level was constantly below the ground surface (Figure 7). The WTD was higher in 2007, 2011–2012, and 2015–2016 in the transect than those in other periods. pH was higher at the 0–10 m BGT transect 1 sites and lower at the 60–70 m BGT transect 1 sites than those at other BGT transect 1 sites and it fluctuated. EC was higher in the 0–20 m BGT transect 1 sites than those at other BGT transect 1 sites and was the highest in 2011. $Ca^{2+}$ concentration was higher at the 0–20 m BGT transect 1 sites than those at other BGT transect 1 sites. Furthermore, $Ca^{2+}$ concentrations were higher in 2010 and 2016 than those in other periods. $SO_4^{2-}$ concentrations were higher at the 0–20 m and 70 m BGT transect 1 sites than those at other BGT transect 1 sites. Furthermore, $SO_4^{2-}$ concentrations were higher in 2007, 2010, and 2016 than those in other periods. TOC was higher at the 0, 40, and 70 m BGT transect 1 sites than those at other BGT transect 1 sites, and $SO_4^{2-}$ concentrations were higher in 2006, 2008, and 2012 than those in other periods.

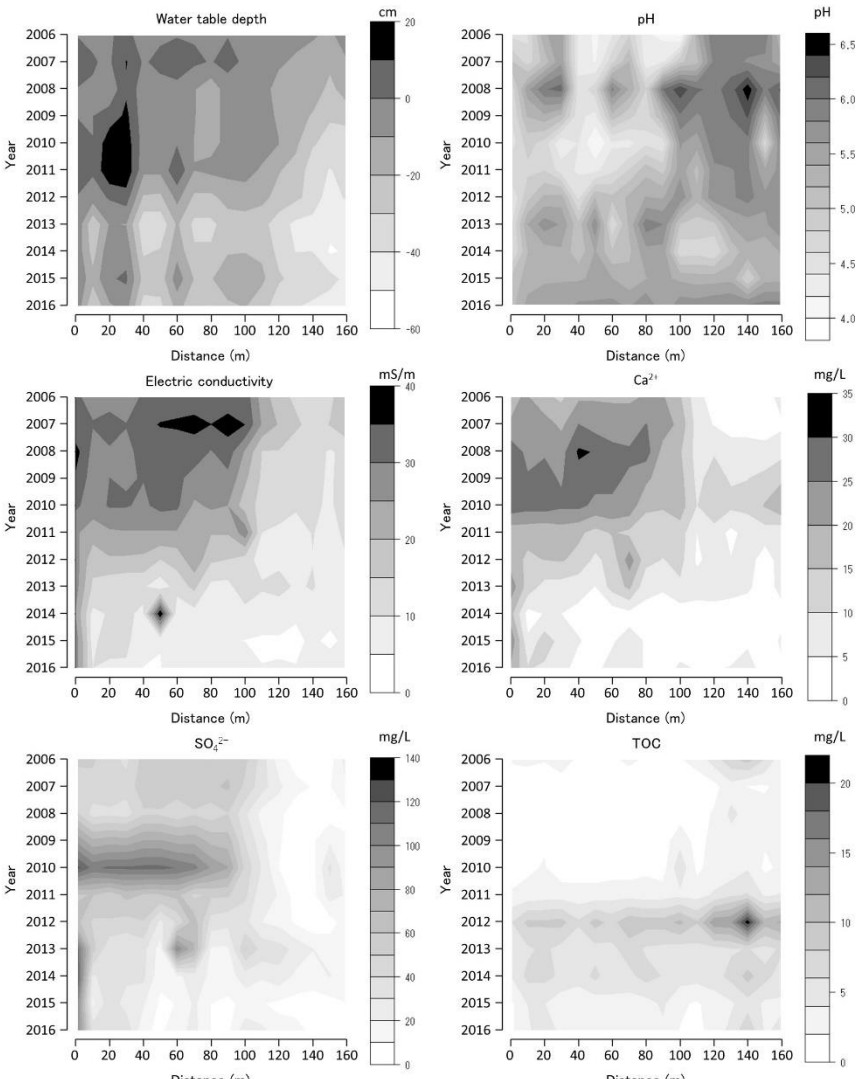

**Figure 6.** Water-table depth (WTD), pH, electric conductivity (EC), calcium ion concentration (Ca²⁺), sulfate ion concentration ($SO_4^{2-}$), and total organic carbon (TOC) along the Tadewara transect (TDW transect). Data were collected at 17 sites of water environment monitoring with 10 m intervals along the transect. Countered figures of chemical variables on the axes of year and position were drawn by interpolating missing data (in 2009).

The WTD was high at the 80–90 m BGT transect 2 sites and were constantly inundated (Figure 8). The WTD in the BGT transect 2 decreased from 2006 to 2008, increased from 2008 to 2012, and decreased from 2012 to 2016. The 60 m BGT transect 2 site was inundated between 2010 and 2013. The pH was higher at the 20–80 m BGT transect 2 sites than those at other BGT transect 2 sites, and higher pH values were observed in 2006, 2010, 2013, and 2016 than those in other periods. The EC was higher at the 0–30 and 70–90 m BGT transect 2 sites and was almost constant from 2006 to 2016. Ca²⁺ concentration was high at the 0–30 and 70–90 m BGT transect 2 sites and was higher in 2006, 2012, and 2016 than those in other periods. $SO_4^{2-}$ concentration was high at the 0–40 and 70–90 m sites of the BGT transect 2 and was higher in 2006, 2012, and 2014–2016 than those in other periods. TOC was higher at the 30–60 m BGT transect 2 sites in 2006 than those of other BGT transect 2 sites and was the highest in 2012. TOC decreased from 2012 to 2016, except at the 70 m BGT transect 2 site.

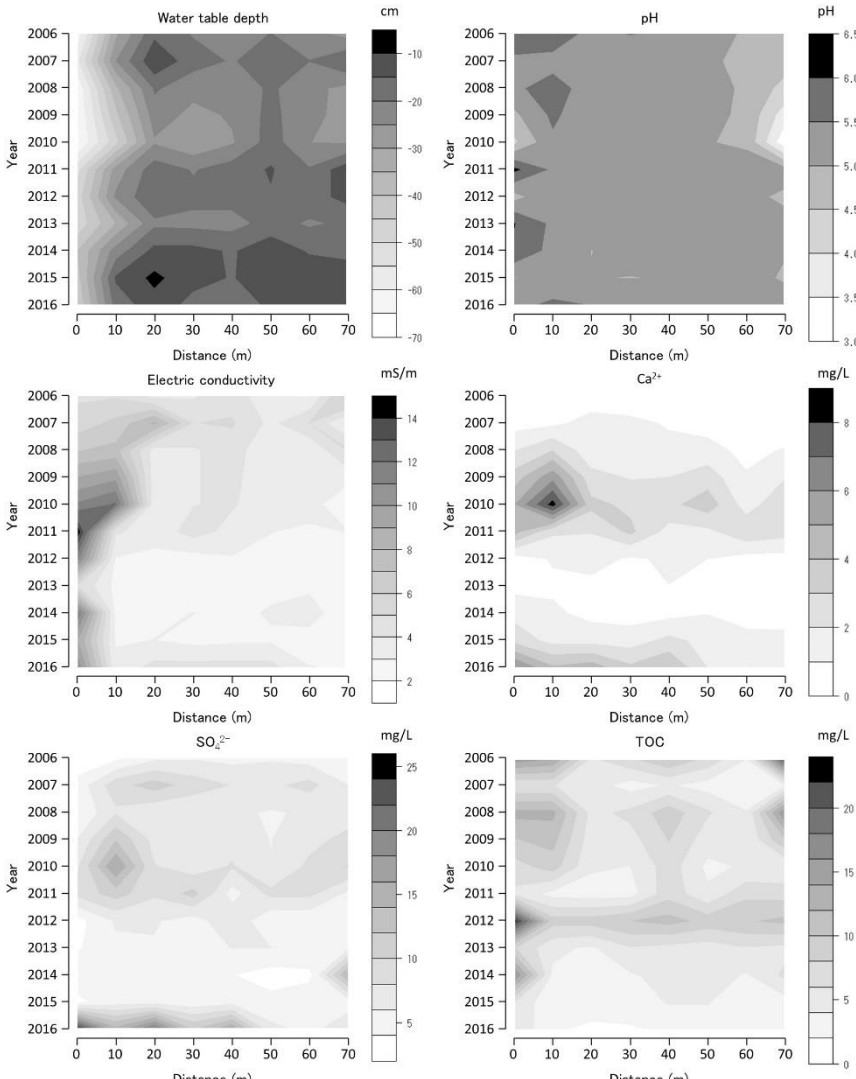

**Figure 7.** Water-table depth (WTD), pH, electric conductivity (EC), calcium ion concentration (Ca$^{2+}$), sulfate ion concentration (SO$_4{}^{2-}$), and total organic carbon (TOC) along Bougatsuru transect 1 (BGT transect 1). Data were collected at 8 sites of water environment monitoring with 10 m intervals along the transect. Countered figures of chemical variables on the axes of year and position were drawn by interpolating missing data (in 2008 and 2009).

### 3.4. Significant Changes in Vegetation

Significant changes in the coverage of abundant species were observed in the TDW transect between 2006 and 2011 (Table 1). The *S. fimbriatum* coverage significantly increased at one site and significantly decreased at two sites. The *S. palustre* and *J. decipiens* coverage significantly increased at four sites and one site, respectively. The *M. japonica*, *P. australis*, and *H. paniculata* coverage significantly decreased at 12, 5, and 7 sites, respectively.

Significant changes in the coverage of abundant species were observed in the TDW transect between 2011 and 2016 (Table 1). The *S. fimbriatum* coverage decreased at six sites, whereas that of *M. japonica*, *P. australis*, and *J. decipiens* significantly increased at 14, 1, and 2 sites, respectively. The *S. palustre* coverage significantly increased at two sites and decreased at one site. The *H. paniculata* coverage significantly increased at one site and decreased at one site.

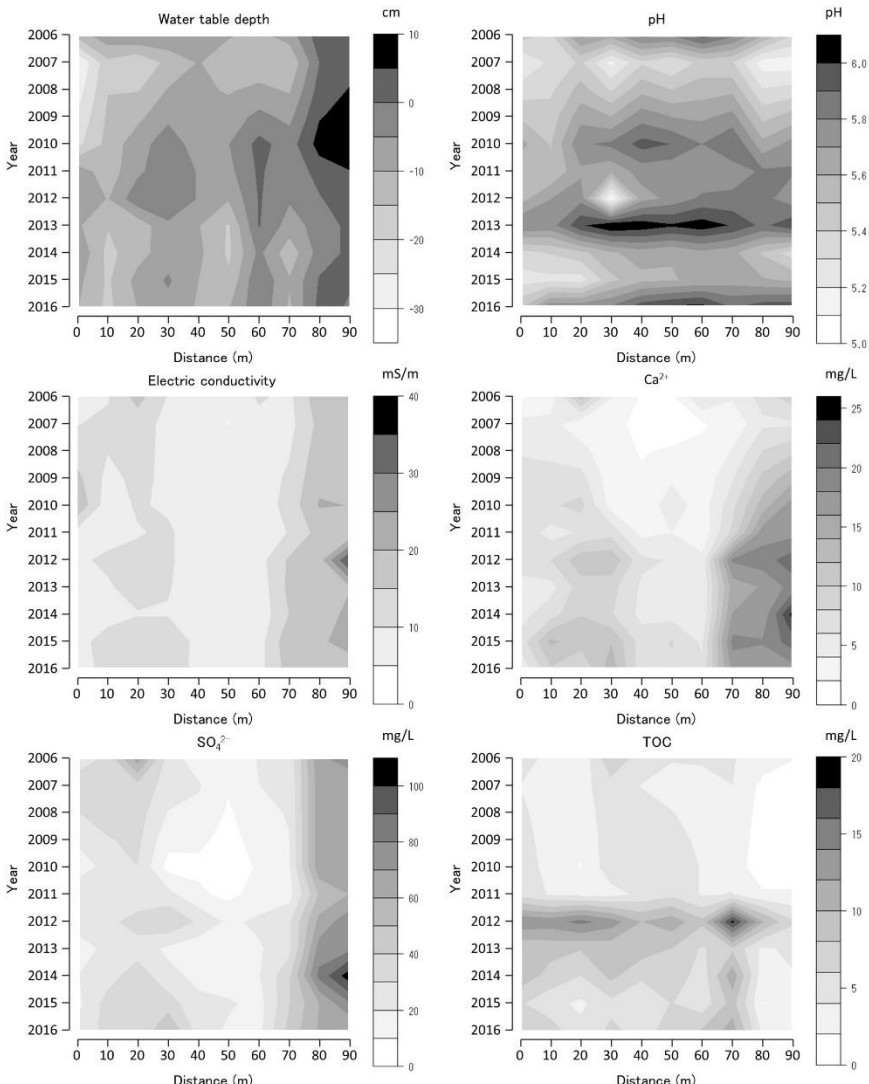

**Figure 8.** Water-table depth (WTD), pH, electric conductivity (EC), calcium ion concentration (Ca²⁺), sulfate ion concentration (SO₄²⁻), and total organic carbon (TOC) along Bougatsuru transect 2 (BGT transect 2). Data were collected at 10 sites of water environment monitoring with 10 m intervals along the transect. Countered figures of chemical variables on the axes of year and position were drawn by interpolating missing data (in 2008 and 2009).

Among the abundant species on BGT transect 1, the *S. fimbriatum* and *S. palustre* coverage significantly decreased from 2006 to 2011 at three sites, whereas that of *P. australis* and *J. decipiens* significantly increased at one and two sites, respectively (Table 2). The *M. japonica* coverage significantly decreased at three sites and increased at one site from 2006 to 2011. From 2011 to 2016, the *S. fimbriatum* and *S. palustre* coverage significantly decreased at one and two sites, respectively, whereas that of *P. australis* and *J. decipiens* significantly increased at two and one sites, respectively. The coverage of *M. japonica* decreased significantly at four sites and increased at one site from 2011 to 2016.

Among the abundant species on BGT transect 2, the *S. fimbriatum* coverage significantly decreased from 2006 to 2011 at five sites, whereas that of *J. decipiens* and *P. thunbergii* increased at one and two sites, respectively (Table 3). The *S. palustre* and *M. japonica* coverage significantly decreased at five and six sites, respectively, from 2006 to 2011, whereas it increased at the 0 and 90 m BGT transect 2 sites. From 2011 to 2016, *S. fimbriatum*, *S. palustre*, *M. japonica*, *J. decipiens*, and *P. thunbergii* coverage significantly increased at one,

two, two, one, and one site, respectively. The *P. australis* coverage significantly increased at three sites and decreased at one site from 2011 to 2016.

**Table 1.** Significance levels of difference of coverage of abundant species between 2006 and 2011 and between 2011 and 2016 at water environment monitoring sites along the Tadewara transect (TDW transect). Five $1 \times 1$ m$^2$ quadrats of both sides of each water environment monitoring site (one side at both end sites) were assigned to each site. Significance of difference was tested by Wilcoxon test. Red-colored cells show the increase in coverage and blue-colored cells show the decrease in coverage from the beginning to the end of each period.

**Tadewara Mire from 2006 to 2011**

| Distance (m) | 0 | 10 | 20 | 30 | 40 | 50 | 60 | 70 | 80 | 90 | 100 | 110 | 120 | 130 | 140 | 150 | 160 |
|---|---|---|---|---|---|---|---|---|---|---|---|---|---|---|---|---|---|
| *Sphagnumfimbriatum* | | | | | | | | * | * | | | | | * | | | |
| *Sphagnumpalustre* | | | | | | | | | ** | | | | * | ** | * | | |
| *Moliniopsisjaponica* | | | * | ** | ** | ** | ** | ** | ** | ** | ** | | | * | * | * | |
| *Phragmitesaustralis* | | | ** | * | | | | | | | * | | * | | | * | |
| *Juncusdecipiens* | | * | | | | | | | | | | | | | | | |
| *Hydrangeapaniculata* | | | | | | | | | * | * | * | ** | * | * | | * | |

**Tadewara Mire from 2011 to 2016**

| Distance (m) | 0 | 10 | 20 | 30 | 40 | 50 | 60 | 70 | 80 | 90 | 100 | 110 | 120 | 130 | 140 | 150 | 160 |
|---|---|---|---|---|---|---|---|---|---|---|---|---|---|---|---|---|---|
| *Sphagnumfimbriatum* | | | | | * | ** | * | * | * | | | | | | ** | | |
| *Sphagnumpalustre* | | | * | | | ** | | | | | | | | | | * | |
| *Moliniopsisjaponica* | * | ** | * | * | * | | ** | | * | * | ** | * | | * | * | * | * |
| *Phragmitesaustralis* | | | ** | | | | | | | | | | | | | | |
| *Juncusdecipiens* | | | | | | | | | | | | | * | * | | | |
| *Hydrangeapaniculata* | * | | | | | | | | | | | * | | | | | |

Note(s): **: $p < 0.01$, *: $p < 0.05$, blank: not significant.

**Table 2.** Significance levels of difference of coverage of abundant species between 2006 and 2011 and between 2011 and 2016 at water environment monitoring sites along Bougatsuru transect 1 (Bougatsuru transect 1). Five $1 \times 1$ m$^2$ quadrats of both sides of each water environment monitoring site (one side at both end sites) were assigned to each site. Significance of difference was tested by Wilcoxon test. Red-colored cells show the increase in coverage and blue-colored cells show the decrease in coverage from the beginning to the end of each period.

**Bougatsuru Mire Transect 1 from 2006 to 2011**

| Distance (m) | 0 | 10 | 20 | 30 | 40 | 50 | 60 | 70 |
|---|---|---|---|---|---|---|---|---|
| *Sphagnumfimbriatum* | | | | * | | ** | * | |
| *Sphagnumpalustre* | | | | | ** | ** | * | |
| *Moliniopsisjaponica* | * | * | * | | | ** | | |
| *Phragmitesaustralis* | | | | | * | | | |
| *Juncusdecipiens* | | * | * | | | | | |

**Bougatsuru Mire Transect 1 from 2011 to 2016**

| Distance (m) | 0 | 10 | 20 | 30 | 40 | 50 | 60 | 70 |
|---|---|---|---|---|---|---|---|---|
| *Sphagnumfimbriatum* | | | * | | | | | |
| *Sphagnumpalustre* | | | | * | | * | | |
| *Moliniopsisjaponica* | | * | | * | * | ** | * | |
| *Phragmitesaustralis* | | * | * | | | | | |
| *Juncusdecipiens* | * | | | | | | | |

Note(s): **: $p < 0.01$, *: $p < 0.05$, blank: not significant.

**Table 3.** Significance levels of difference of coverage of abundant species between 2006 and 2011 and between 2011 and 2016 at water environment monitoring sites along the Bougatsuru transect 2 (Bougatsuru transect 2). Five $1 \times 1$ m$^2$ quadrats of both sides of each water environment monitoring site (one side at both end sites) were assigned to each site. Significance of difference was tested by Wilcoxon test. Red colored cells show the increase in coverage and blue colored cells show decrease in coverage from the beginning to the end of each period.

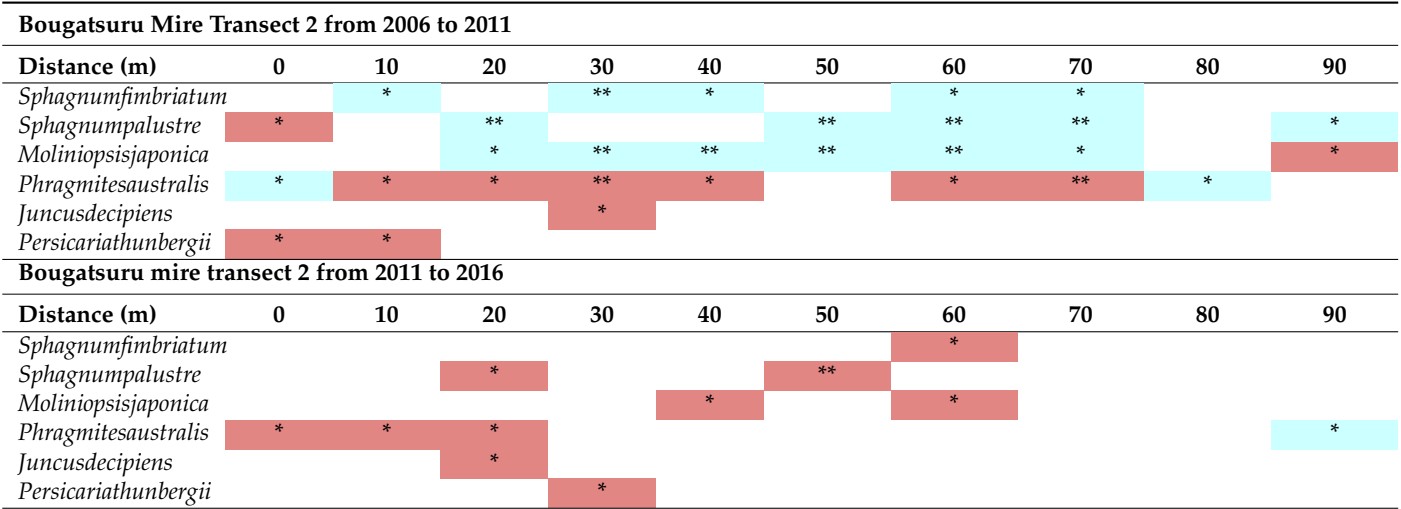

Note(s): **: $p < 0.01$, *: $p < 0.05$, blank: not significant.

### 3.5. Significant Changes in the Water Environment

Significant changes in WTD were observed at the TDW transect from 2006 to 2011 (Table 4). The WTD significantly increased at four sites and decreased at 11 sites. The pH significantly increased at both sites and decreased at all three sites. The EC and Cl$^-$, Na$^+$, K$^+$, and Mg$^{2+}$ concentrations significantly decreased at 15, 15, 15, 12, and 16 sites, respectively, whereas NO$_3^-$, NH$_4^+$, TOC, TP, and TN concentrations significantly increased at 4, 2, 8, 16, and 3 sites, respectively. The SO$_4^{2-}$ concentration significantly increased at five sites and decreased at one site. Ca$^{2+}$ concentration significantly increased at three sites and decreased at four sites.

Significant changes in WTD, EC, and Cl$^-$, Ca$^{2+}$, TOC, and TN concentrations were observed at the TDW transect from 2011 to 2016 (Table 4). WTD, EC, and Cl$^-$, Ca$^{2+}$, TOC, and TN concentrations significantly decreased at 12, 14, 15, 10, 10, and 3 sites, respectively, whereas pH, NO$_3^-$, and TP significantly increased at 8, 16, and 4 sites, respectively. The SO$_4^{2-}$, Na$^+$, K$^+$, and Mg$^{2+}$ concentrations significantly decreased at 10, 12, 6, and 8 sites, respectively, whereas they increased at the 0 m TDW transect site. The NH$_4^+$ concentration increased significantly at one site and significantly decreased at one site.

Significant changes in WTD, NO$_3^-$, and TP concentrations were observed at the TDW transect (Table 5). Data from 2011 to 2016 are shown because of the scarcity of data from 2006 to 2011. The WTD, and NO$_3^-$, and TP concentrations significantly increased at three, seven, and one site, respectively, whereas pH and Cl$^-$, NH$_4^+$, Na$^+$, K$^+$, and TOC concentrations decreased at one, two, one, one, three, and seven sites, respectively.

Significant changes were observed in the BGT transect 2 (Table 6). Data from 2011 to 2016 are shown because of the scarcity of data from 2006 to 2011. The Na$^+$, K$^+$, and Mg$^{2+}$ concentrations in spring water significantly increased, whereas TOC decreased. WTD, pH, Cl$^-$, and NH$_4^+$ concentrations in groundwater significantly decreased at five, one, nine, and three sites, respectively, whereas EC, NO$_3^-$, SO$_4^{2-}$, Na$^+$, and Mg$^{2+}$ concentrations significantly increased at six, four, four, five, and nine sites, respectively. The K$^+$ and Ca$^{2+}$ concentrations of groundwater significantly increased at two and six sites, respectively, whereas they decreased at the 0 m site. The TN concentration of groundwater significantly

increased at the 0 m site and significantly decreased at the 20 m site in BGT transect 2 from 2011 to 2016.

**Table 4.** Significance levels of regression of water environmental variables and elapsed days in Tadewara transect (TDW transect) from 2006 to 2011 and from 2011 to 2016. Red-colored cells mean significant positive slopes and blue-colored cells mean significant negative slopes.

**Tadewara Mire from 2006 to 2011**

| Distance (m) | 0 | 10 | 20 | 30 | 40 | 50 | 60 | 70 | 80 | 90 | 100 | 110 | 120 | 130 | 140 | 150 | 160 |
|---|---|---|---|---|---|---|---|---|---|---|---|---|---|---|---|---|---|
| WTD | ** |  | ** | ** | ** | ** | * | ** | * | ** |  | * | * | ** | ** | ** | ** |
| pH |  |  | * | *** |  |  |  |  | ** |  | ** |  |  |  |  | ** |  |
| EC | * | ** | ** |  | *** | *** | *** | *** | *** | *** |  | *** | *** | *** | * | ** | * |
| Cl⁻ | ** | ** | *** | ** | ** | *** | *** | *** | *** | *** | ** |  | ** | ** | *** | *** | *** |
| NO₃⁻ | *** |  |  |  | * |  |  |  |  |  |  |  |  |  |  | ** | * |
| SO₄²⁻ | * |  |  | *** | * | * |  |  |  |  |  | * |  |  |  | * |  |
| NH₄⁺ |  |  |  |  |  |  |  |  |  |  |  |  | ** |  | * |  |  |
| Na⁺ | * | ** | ** | * | ** | ** | ** | ** | *** | *** | ** |  |  | *** | *** | *** | *** |
| K⁺ |  | * | ** | * |  | ** | *** |  | ** | *** |  |  | * | * | *** | ** | *** |
| Mg²⁺ | ** | *** | *** | *** | *** | *** | *** | *** | *** | *** | *** | *** | ** | ** |  | * | ** |
| Ca²⁺ |  |  |  |  |  |  | * |  | *** | *** | * |  | * |  | *** | ** |  |
| TOC |  | * |  |  | ** | ** | ** | ** | ** | *** |  | * |  |  |  |  |  |
| TP | *** | ** | ** | ** | ** | * | ** | * | *** | *** | * | ** | ** | ** | ** |  | *** |
| TN |  | * |  |  |  |  |  |  | * | * |  |  |  |  |  |  |  |

**Tadewara Mire from 2011 to 2016**

| Distance (m) | 0 | 10 | 20 | 30 | 40 | 50 | 60 | 70 | 80 | 90 | 100 | 110 | 120 | 130 | 140 | 150 | 160 |
|---|---|---|---|---|---|---|---|---|---|---|---|---|---|---|---|---|---|
| WTD |  | ** | ** | ** | ** | *** | ** | ** | ** | ** |  | ** | ** | * |  |  |  |
| pH |  |  | *** | *** | *** | *** | *** | ** | ** |  |  | * |  |  |  |  |  |
| EC |  | *** | *** | *** | *** |  | *** | *** | *** | *** | *** | *** | *** | *** | ** |  | ** |
| Cl⁻ |  | *** | * | *** | *** | *** | *** | *** | *** | *** | *** | *** | *** |  | *** | *** | *** |
| NO₃⁻ | *** | ** | * | ** |  | ** | ** | * | ** | ** | * | ** | ** | ** | ** | ** | ** |
| SO₄²⁻ | ** | ** |  | * | *** | *** |  | *** | *** | *** | * |  |  |  |  | * | ** |
| NH₄⁺ |  |  |  |  |  |  | * |  |  |  |  |  |  |  |  | * |  |
| Na⁺ | ** | ** |  | ** | *** | *** | *** | *** | *** | *** | *** | * | * |  |  | ** |  |
| K⁺ | * |  |  |  |  | ** |  |  | * |  | * |  | * | * |  | * |  |
| Mg²⁺ | ** |  |  | ** | *** | *** | ** | *** | *** | *** | ** |  |  |  |  |  |  |
| Ca²⁺ |  | ** |  | *** | *** | *** | ** | *** | *** | *** | ** |  | * |  |  |  |  |
| TOC | ** |  |  | * | * |  |  | * |  |  | *** | *** | *** | ** |  | ** | * |
| TP |  | ** | * | ** |  | * |  |  |  |  |  |  |  |  |  |  |  |
| TN |  |  |  |  |  |  |  | ** |  |  | * |  |  |  |  |  | * |

Note(s): ***: $p < 0.001$, **: $p < 0.01$, *: $p < 0.05$, blank: not significant.

**Table 5.** Significance levels of regression of water environmental variables and elapsed days in Bougatsuru transect 1 (BGT transect 1) from 2011 to 2016. Red-colored cells mean significant positive slopes and blue-colored cells mean significant negative slopes.

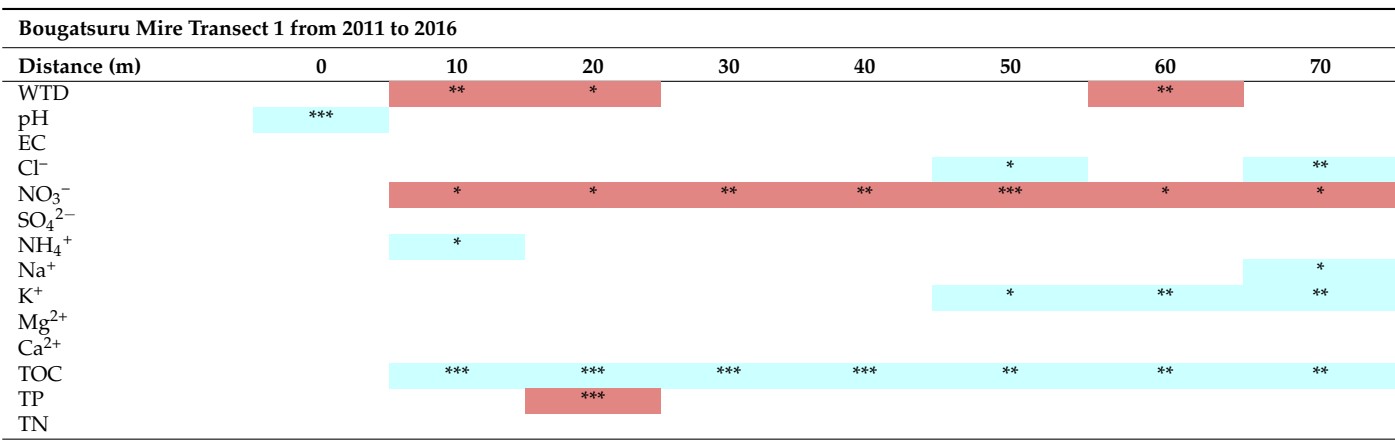

**Bougatsuru Mire Transect 1 from 2011 to 2016**

| Distance (m) | 0 | 10 | 20 | 30 | 40 | 50 | 60 | 70 |
|---|---|---|---|---|---|---|---|---|
| WTD |  | ** | * |  |  |  | ** |  |
| pH | *** |  |  |  |  |  |  |  |
| EC |  |  |  |  |  |  |  |  |
| Cl⁻ |  |  |  |  |  | * |  | ** |
| NO₃⁻ |  | * | * | ** | ** | *** | * | * |
| SO₄²⁻ |  |  |  |  |  |  |  |  |
| NH₄⁺ |  | * |  |  |  |  |  |  |
| Na⁺ |  |  |  |  |  |  |  | * |
| K⁺ |  |  |  |  |  | * | ** | ** |
| Mg²⁺ |  |  |  |  |  |  |  |  |
| Ca²⁺ |  |  |  |  |  |  |  |  |
| TOC |  | *** | *** | *** | *** | ** | ** | ** |
| TP |  |  | *** |  |  |  |  |  |
| TN |  |  |  |  |  |  |  |  |

Note(s): ***: $p < 0.001$, **: $p < 0.01$, *: $p < 0.05$, blank: not significant.

**Table 6.** Significance levels of regression of water environmental variables and elapsed days in Bougatsuru transect 2 (BGT transect 2) from 2011 to 2016. Red-colored cells mean significant positive slopes and blue-colored cells mean significant negative slopes.

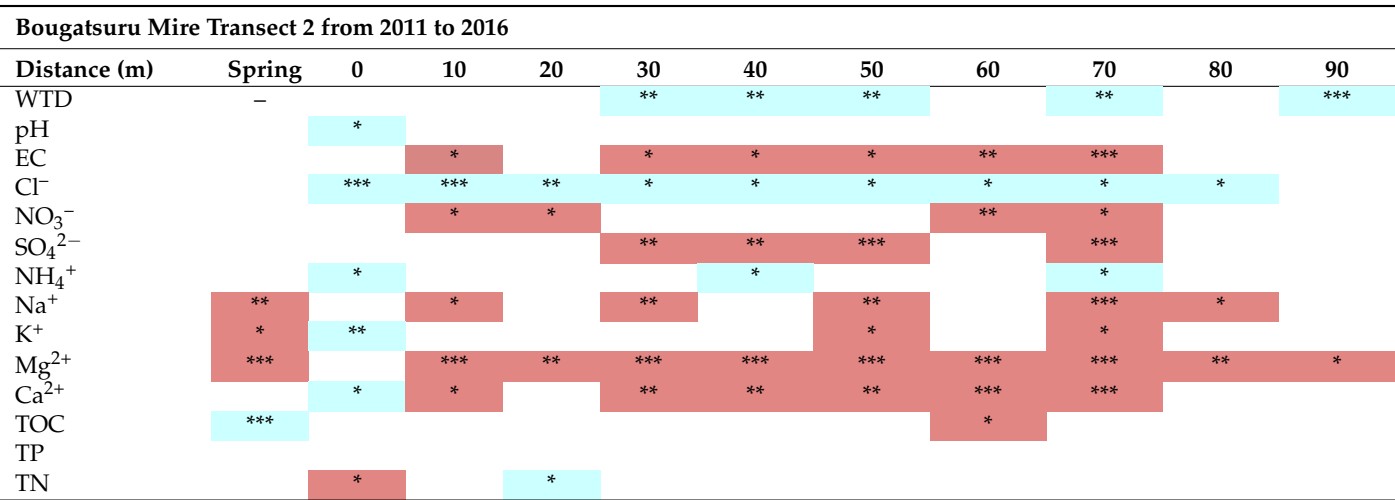

| Bougatsuru Mire Transect 2 from 2011 to 2016 | | | | | | | | | | | |
|---|---|---|---|---|---|---|---|---|---|---|---|
| Distance (m) | Spring | 0 | 10 | 20 | 30 | 40 | 50 | 60 | 70 | 80 | 90 |
| WTD | – | | | | ** | ** | ** | | ** | | *** |
| pH | | * | | | | | | | | | |
| EC | | | * | | * | * | * | ** | *** | | |
| $Cl^-$ | | *** | *** | ** | * | * | * | * | * | * | |
| $NO_3^-$ | | | * | * | | | | ** | * | | |
| $SO_4^{2-}$ | | | | | ** | ** | *** | | *** | | |
| $NH_4^+$ | | * | | | | * | | | * | | |
| $Na^+$ | ** | | * | | ** | | ** | | *** | * | |
| $K^+$ | * | ** | | | | | * | | * | | |
| $Mg^{2+}$ | *** | | *** | ** | *** | *** | *** | *** | *** | ** | * |
| $Ca^{2+}$ | | * | * | | ** | ** | ** | *** | *** | | |
| TOC | *** | | | | | | | * | | | |
| TP | | | | | | | | | | | |
| TN | | * | | * | | | | | | | |

Note(s): ***: $p < 0.001$, **: $p < 0.01$, *: $p < 0.05$, blank: not significant.

### 3.6. Correlation between Vegetation and Water Environmental Change

A significant positive correlation was observed between the *S. fimbriatum* coverage and $Ca^{2+}$ concentration; *M. japonica* coverage and either pH or $Mg^{2+}$ concentration; *P. australis* coverage and $Na^+$ or $Mg^{2+}$ concentration; and *H. paniculata* coverage and either $Na^+$, $K^+$, or $Mg^{2+}$ concentration (Table 7). A significant negative correlation was observed between the *M. japonica* coverage and $SO_4^{2-}$, TOC, and TN concentrations, and the *H. paniculata* coverage and TN concentration.

**Table 7.** Significance levels of correlation between coverage of each abundant species and water environmental variables in transects of the TDW transect, BGT transect 1, and BGT transect 2. Significance levels of Spearman's' correlation coefficient were shown. Red-colored cells mean significant positive correlations and blue-colored cells mean significant negative correlations.

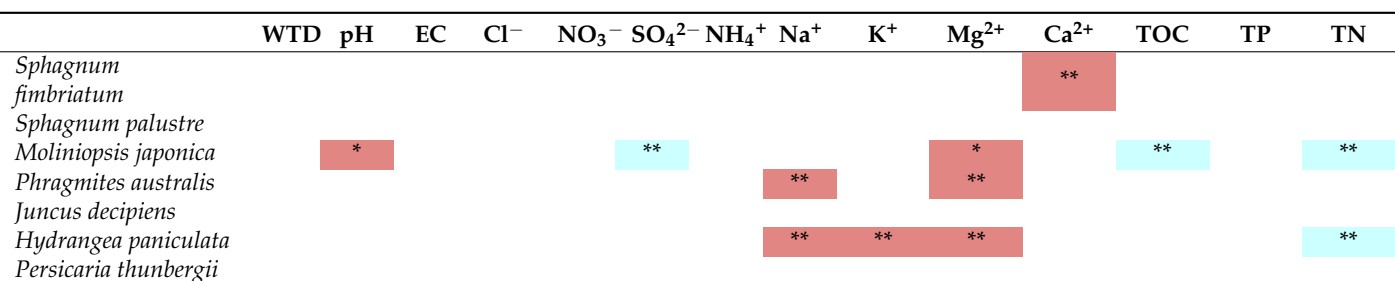

| | WTD | pH | EC | $Cl^-$ | $NO_3^-$ | $SO_4^{2-}$ | $NH_4^+$ | $Na^+$ | $K^+$ | $Mg^{2+}$ | $Ca^{2+}$ | TOC | TP | TN |
|---|---|---|---|---|---|---|---|---|---|---|---|---|---|---|
| *Sphagnum fimbriatum* | | | | | | | | | | | ** | | | |
| *Sphagnum palustre* | | | | | | | | | | | | | | |
| *Moliniopsis japonica* | | * | | | | ** | | | | * | | ** | | ** |
| *Phragmites australis* | | | | | | | | ** | | ** | | | | |
| *Juncus decipiens* | | | | | | | | | | | | | | |
| *Hydrangea paniculata* | | | | | | ** | | ** | ** | | | | | ** |
| *Persicaria thunbergii* | | | | | | | | | | | | | | |

Note(s): **: $p < 0.01$, *: $p < 0.05$, blank: not significant.

## 4. Discussion

### 4.1. Evidence of Changes in the Vegetation and Water Environment

Monthly average atmospheric temperature and monthly precipitation in Kusu Meteorological Observatory located near the Tadewara mire and the Bougatsuru mire (33.292° N, 131.155° E, 331 m a.s.l.; https://www.data.jma.go.jp/obd/stats/etrn/index.php?prec_no=83&block_no=0932&year=2022&month=&day=&view=) accessed on 13 November 2022 showed no significant secular change from 2006 to 2016. Eruption or a discernible increase in volcanic earthquake and volcanic smoke height from fumarole were never reported in Kujusan (nearest observation station of volcano from the investigation sites) from 2006 to 2016 (https://www.data.jma.go.jp/vois/data/anicul/502_Kujusan/502_index.html ac-

cessed on 13 November 2022). Therefore, direct effects of climate change and volcanic activity on vegetation in the investigated mires are regarded to be limited.

The coverage of every abundant species and its pattern of change on each transect was heterogeneous. Vegetation on the TDW transect in 2006 was dominated by *S. palustre*, *S. fimbriatum*, and *M. japonica* (Figure 3). *P. australis* and *H. paniculate* were the dominant species at some sites on the TDW transect. The *Sphagnum* spp. coverage decreased from 2006 to 2010 and then increased until 2012. The *M. japonica* coverage increased from 2006 to 2008 and decreased until 2012. The high *Sphagnum* spp. coverage corresponded to the low *M. japonica* coverage during the 2011–2012 period. The WTD increased and EC decreased from 2006 to 2012 (Figure 6). The $Ca^{2+}$ concentration decreased from 2010, corresponding to the temporal $SO_4^{2-}$ increase and a pH decrease in 2010. Inundated water at the 0–100 m TDW transect sites supplies sulfuric acid, and $Ca^{2+}$ would be removed through $CaSO_4$ precipitation. Therefore, the *Sphagnum* spp. coverage increased during the 2010–2012 period because of growth inhibition owing to the removal of $Ca^{2+}$ [14]. *Sphagnum*-dominated vegetation changed to *M. japonica*-dominated vegetation from 2012 to 2016 (Figure 3). WTD decreased from 2012 to 2016, and the dryness of the soil surface increased, leading to a decline in *Sphagnum* coverage and an increase in *M. japonica* coverage. WTD was the lowest during the 2012–2014 period, corresponding to the high TOC owing to the increased decomposition of organic materials in the soil. The TDW transect vegetation changed from *Sphagnum-M. japonica*-dominated vegetation to *Sphagnum*-dominated vegetation via *M. japonica* dominated vegetation from 2006 to 2012. Acidification and $Ca^{2+}$ removal by an inundation of sulfuric acid-contaminated water affected by volcanic activity changed the vegetation from *M. japonica*-dominated vegetation to *Sphagnum*-dominated vegetation from 2006 to 2012. The increase in dryness was caused by a decrease in the flow of the neighboring stream, which changed vegetation from *Sphagnum*-dominated to *M. japonica*-dominated vegetation from 2012 to 2016.

The *Sphagnum* spp. coverage in BGT transect 1 decreased temporarily in 2010–2011 (with some exceptions), whereas that of *M. japonica* increased during the same period. During the 2010–2011 period, WTD decreased temporarily, and $Ca^{2+}$ and $SO_4^{2-}$ concentrations temporarily increased (Figure 7). The effect of sulfuric acid contamination on groundwater in the BGT transect 1 was significant. Acidification by sulfuric acid promoted $Ca^{2+}$ dissolution from the bedrock scattered on the surface of BGT transect 1, causing a decrease in the *Sphagnum* spp. coverage during this period. A high TOC concentration was observed after the WTD decline during the 2008–2010 period and in 2012, implying that soil organic matter decomposition increased owing to the increased oxygen supply due to the decline in WTD. The BGT transect 1 vegetation temporarily changed from *Sphagnum*-dominated to *M. japonica*-dominated vegetation, due to a temporal increase in $Ca^{2+}$ and $SO_4^{2-}$ concentrations owing to volcanic activity; however, by the end of the study period the *Sphagnum* vegetation had recovered (2016).

The *Sphagnum* spp. coverage in 0–70 m at BGT transect 2 sites (except for the upper part (0–20 m sites) of the transect) decreased after 2011. Contrastingly, $Ca^{2+}$ concentration increased, causing a decline in *Sphagnum* coverage (Figure 8). The *P. australis* coverage increased, corresponding to a decrease in *Sphagnum* coverage. The inundated sites with high water tables (75–90 m sites) had increased *M. japonica* and *J. decipiens* coverage corresponding to an increase in $Ca^{2+}$ and $SO_4^{2-}$ concentrations. The inundated water flowed from the spring next to the 0 m site; however, the spring water flowed to the edge of and into the mire at the lower part (75–90 m sites) of BGT transect 2. The $Ca^{2+}$ concentration in spring water increased from 2011 to 2016 (data are presented in supplementary materials) owing to the increase in the $SO_4^{2-}$ concentration. The BGT transect 2 vegetation changed from *Sphagnum*-dominated to *P. australis*-dominated on the upper part of the slope owing to increased $Ca^{2+}$ and $SO_4^{2-}$ concentrations in flowing water after volcanic activity.

*4.2. Effect of the Water Environment on Vegetation Change*

TDW and BGT transects located under the knick points of the mountainous areas and mire vegetation are fed by spring water flowing out from the knick points. Streams originating from springs flow directly or indirectly into the mires, affecting the groundwater table and quality within them. The groundwater chemistry in mountainous regions is affected by volcanic activity. High sulfuric acid and carbonate concentrations increase groundwater acidity, resulting in high concentrations of dissolved alkaline and alkaline earth metals in groundwater after rock chemical weathering [15]. Low pH and high $Ca^{2+}$ and $Na^+$ concentrations in stream water at the western end of the TDW transect (0 m site) and that in the transect 2 revealed that water from these streams was directly affected by volcanic activity.

Four streams flow into the TDW transect; two flow into the western end (0 m site) and eastern end (160 m site) of the transect line of the TDW transect. The water flow at the western end of the transect was constant throughout the study period (2006–2016). The water table increased from 2006 to 2015 and decreased from 2015 to 2016 (Figure 6; 0 m site). The branch of the stream temporarily appeared in 2010–2011 across the 20 m site of the transect line, which seemed to be a temporal increase in the surface water level at this site (Figure 6). These streams are the primary water sources for the TDW transect vegetation. Streams at the eastern end of the transect line were sedimented after heavy rainfall in 2007, rapidly decreasing the groundwater level at the 130–160 m sites from 2007 to 2008 (Figure 6). Furthermore, the flow did not recover until the end of the study. The groundwater level near the eastern end of the transect line (160 m site) was directly affected by precipitation. In the TDW transect, the effect of stream water at the 0 m site was similar to that at the 0–80 m sites, and the amount of supplied water increased from 2006 to 2011 and then decreased after 2011. The amount of $Ca^{2+}$ available to the vegetation changed, considering the water available.

BGT transect 1 was on a gentle slope, with no streams at the upper side of the transect line. The groundwater level of the BGT transect 1 sites fluctuated because of precipitation (Figure 7). The 60 m BGT transect 1 site was on a terrace 1 m high and had a pool near the 60 m site. The groundwater table at the 60 m BGT transect 1 site was constant compared with those at the other BGT transect 1 sites. There was no direct water supply from streams to BGT transect 1. Precipitation was the main water source and caused fluctuations in the groundwater-table depth. Changes in the groundwater table affect the redox properties, the decomposition rate of organic materials in soil, and the weathering of the bedrock, consequently affecting water chemistry and vegetation.

BGT transect 2 was on a gentle slope just below the spring of the knick point. Spring water flowed approximately 20 m upstream of the 0 m BGT transect 2 site. The stream flows out of BGT transect 2; the groundwater table at the upper sites of BGT transect 2 was not affected by the stream, whereas a branch of the stream appeared on the lower site of the transect line, and the groundwater table increased at the 0–70 m BGT transect 2 site (Figure 8). The $Ca^{2+}$ concentration in the spring water of BGT transect 2 increased from 2010 to 2012, decreased from 2012 to 2014, and increased from 2014 to 2016 (data are presented in the supplementary materials). The change in $Ca^{2+}$ concentration reflects a change in volcanic activity within the spring water aquifer. The change in $Ca^{2+}$ concentration was observed at the 30–90 m BGT transect 2 sites, revealing that the groundwater at the transect lower site was directly affected by spring water. Changes in the amount and chemical variables of stream water at the BGT transect 2 lower sites (80–90 m sites) caused vegetation changes. Water chemistry changes at the upper site of transect 2 were not directly correlated with those of stream water (data are presented in supplementary materials); therefore, the water-table depth and water chemistry at the upper site of transect 2 were directly affected by precipitation, consistent with the findings from transect 1. The type of water provided to determine vegetation type was different between the upper and lower BGT transect 2 sites.

### 4.3. Vegetation and Water Environment of Mires

The vegetation of mires in volcanic areas is affected by volcanic activity, particularly volcanic ash deposition. Volcanic ash forms a less-permeable tephra layer on vegetation and causes hydrological changes in mires. The less-permeable tephra layer also forms enriched layers by accumulating macro, trace, and toxic elements; thus, the chemical properties of the peat layer are typical of the tephra layer [16–18].

Crowley et al. [19] reported that volcanic ash deposition increases pH and sulfur compounds in the peat layer. Wolejko and Ito [10] noted that tephra supplies nutrients to mire vegetation; however, Hotes et al. [20] revealed that tephra deposition buries vegetation. Nonetheless, the chemical effects of ash deposition on mire vegetation are unknown.

An analysis of macrofossils in peat in the TDW transect revealed that six interfaces of layers with dominant peat-forming species were included within 138 cm of the peat layer from the surface, which dates to $970 \pm 40$ y.B.P. [12]. The dominant species within the peat-forming plant community had frequent variations from the bottom to the surface sediment layer. The vegetation in the TDW transect frequently changed among the ombrogenous (*S. palustre* and *S. fimbriatum*), minerogenous (*P. australis*), and intermediate communities (*M. japonica*). Succession from ombrogenous to minerogenous and from minerogenous to ombrogenous communities implied frequent changes in the water environment in mires affected by volcanic activity.

Hydrological changes in volcanic deposition through secondary deposition after heavy precipitation in and the surrounding areas of mires change vegetation in the TDW and BGT transects. Changes in groundwater level were common at several sites in the TDW and BGT transects; however, the correlation between changes in groundwater level and the coverage of dominant species was not significant (Table 7).

Changes in the *P. australis* coverage over 5 years were positively correlated with the $Na^+$ and $Mg^{2+}$ concentrations. Changes in the coverage of the shrub species *H. paniculata* (established at minerotrophic sites and requiring a high nutrient supply) were positively correlated with $Na^+$, $K^+$, and $Mg^{2+}$ concentrations. An increase in nutrient concentration accelerates the establishment of shrub species. Change in the *H. paniculata* coverage was negatively correlated with TN, implying that there was high nitrogen uptake after the establishment of the species. Changes in *M. japonica* coverage were significantly positively correlated with changes in pH and $Mg^{2+}$ concentrations. Contrastingly, the coverage was significantly negatively correlated with $SO_4^{2-}$, TOC, and TN concentrations. *M. japonica* colonization requires a nutrient supply and neutralizes the preceding *Sphagnum* habitat. Neutralization accelerates the decomposition of dissolved organic carbon in peat and nitrogen consumption. The decrease in $SO_4^{2-}$ concentration coupled with the increase in *M. japonica* coverage was due to a decrease in groundwater supply caused by volcanic activity and an increase in groundwater pH. A significant positive correlation was observed between *S. fimbriatum* and $Ca^{2+}$ concentration (Table 7). The growth of most *Sphagnum* species was negatively affected by calcareous water. Changes in the vegetation and water environment in the TDW transect from 2006 to 2011 (Figures 3 and 6) could be due to the $Ca^{2+}$ inhibitory effect on *Sphagnum* growth. Monitoring data revealed that $Ca^{2+}$ concentration increased and then decreased during this period and rapidly decreased in 2011. The *S. fimbriatum* coverage increased rapidly, corresponding to a rapid decrease in $Ca^{2+}$ concentration. Thus, $Ca^{2+}$ concentration had a significant positive slope, whereas the *S. fimbriatum* coverage in 2011 was significantly higher than that in 2006 owing to the different analysis methods; however, the vegetation change progressed for less than 5 years. A significant positive correlation between *Sphagnum* coverage and $Ca^{2+}$ concentration was observed for *S. fimbriatum*. Among *Sphagnum* species, *S. fimbriatum* and *S. squarossum* are the species that colonize calcareous habitats [21,22]. *S. fimbriatum* first colonized the sites just after the beginning of $Ca^{2+}$ removal from water, and *S. palustre* colonized after a sufficiently low $Ca^{2+}$ concentration in the groundwater.

## 5. Conclusions

The correlation between changes in the groundwater environment and vegetation in volcanic mires confirmed that the water environment changed during the 10-year study period, and subsequent changes in the dominant species of mire vegetation were observed. The nutritional variables of water supplied to the vegetation changed owing to volcanic activity affecting the dominance of minerotrophic species. Thus, vegetation succession in volcanic mires changed from ombrogenous to minerogenous and from minerogenous to ombrogenous communities. The water environment promoted vegetation change.

**Supplementary Materials:** The following supporting information can be downloaded at: http://doi.org/10.13140/RG.2.2.34825.31848.

**Funding:** This study was supported by Seven-Eleven green funds provided by the Seven-Eleven memorial foundation from 2006 to 2015.

**Data Availability Statement:** The databases used in the investigation have been submitted as an annual report to the fund.

**Acknowledgments:** The databases of water environment and vegetation in the Tadewara mire and the Bougatsuru mire were constructed by members of the laboratory of ecosystem functional analysis, Department of Biology, Faculty of Environmental Engineering, The University of Kitakyushu.

**Conflicts of Interest:** The author declares no conflict of interest.

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
