# Peer review of "A Case Study of a 10-Year Change in the Vegetation and Water Environments of Volcanic Mires in South-Western Japan"

_water, doi:10.3390/w14244132_

Round 1

Reviewer 1 Report

This paper contains a good data, methodology and interpretation, but to improve the quality I suggest some minor correction

Please can you more detail the groundwater level fluctuation during these 10 years (and if possible the quality)

You must add some values of this fluctuation in the abstract and in the text

What about the balance of your chemical data? must be under 10%

Is there any relationship between the surface water/groundwater/vegetation during this period?

Have you an idea about the impact of climate change on surface water and groundwater (you can read the paper of Hamed et al., 2018)

Is there a change in some vegetation spaces vs climate change ?

More detail the relationship between vegetation/quality of water/quality of soil (changing in structure and texture)

Page 11 lines (311-312): The H. paniculata coverage significantly increased at one site and de- 311 creased at one site.  (More explain why?)

Added some recommendations 

Author Response

Reply to the Reviewer #1

Thank you very much for your constructive comments on my manuscript: water-2081383. I have made revision on my manuscript according to your suggestion.

This paper contains a good data, methodology and interpretation, but to improve the quality I suggest some minor correction.

Answer: Thank you very much for evaluating my paper.

Please can you more detail the groundwater level fluctuation during these 10 years (and if possible the quality). You must add some values of this fluctuation in the abstract and in the text

Answer: I think that the detailed data of ground water level fluctuation both in the abstract and in the main text are quite helpful to understand the document as you suggested. However, the changes in water environmental variables are highly heterogeneous among mires and data collecting sites. Therefore, the documentation of the detailed data in the text will make confusion. That is why I used the statistical analysis of water environmental change and selected only significant change. I added the maximum value of Ca2+ in abstract because Ca2+ is the determining factor of Sphagnum growth (l. 14).

What about the balance of your chemical data? must be under10%

Answer: Balance of positive and negative charges in each water sample were checked. Some samples show high charge balance deficit; however, it is due to the high total organic carbon (TOC) concentration. Ground water in peatlands usually show high TOC, therefore I confirm that the determination of ionic concentration has been appropriately made.

Is there any relationship between the surface water/groundwater/vegetation during this period?

Answer: Surface water and groundwater were not separately analyzed because inundation was observed only in limited sites in the specific period. Significant correlation between water environment and vegetation (abundance of dominant species) are summarized in Table 7.

Have you an idea about the impact of climate change on surface water and groundwater (you can read the paper of Hamed et al.,2018)? Is there a change in some vegetation spaces vs climate change?

Answer: Data of atmospheric temperature and annual precipitation provided by Japan Meteorological Agency (https://www.data.jma.go.jp/obd/stats/etrn/index.php) show no significant change in meteorological variables during the investigation period from 2006 to 2016. Therefore, I think that the impact of climate change on water chemistry in the investigation sites is not significant. I added the description in ll. 416–425; 'Monthly average atmospheric temperature and monthly precipitation in Kusu Mete-orological Observatory located near the Tadewara mire and the Bougatsuru mire (33.292°N, 131.155°E, 331 m a.s.l.; https://www.data.jma.go.jp/obd/stats/etrn/index.php?prec_no=83&block_no=0932&year=2022&month=&day=&view=) showed no significant secular change from 2006 to 2016. Eruption or discernible increase of volcanic earthquake and volcanic smoke height from fumarole were never reported in Kujusan (nearest observation station of volcano from the investigation sites) from 2006 to 2016  (https://www.data.jma.go.jp/vois/data/fukuoka/502_Kujusan/502_index.html). Therefore, direct effects of climate change and volcanic activity on vegetation in the investigated mires are regarded to be limited.'

More detail the relationship between vegetation/quality of water/quality of soil (changing in structure and texture).

Answer: I think that the more detailed analysis on the relationship between water chemistry and vegetation is possible by using huge data included in the paper. However, the change of water environment and vegetation are highly heterogeneous among sites. Therefore, I selected the only statistically significant relationship appearing in Table 7 and discuss about them.

Page 11 lines (311-312): The H. paniculata coverage significantly increased at one site and decreased at one site. (More explain why?)

Answer: Hydrangea paniculata is a woody species and the growth of some small number of individuals would affect change in coverage of the species. Decrease of water table would accelerate growth of the species, whereas inundation would inhibit the growth of the species. I think the water table change within limited area (sometimes outside the water table measuring sites) would affect the coverage of H. paniculata. However, this is only a speculation without evidence. So, I gave up discussing about this point and used the data for the correlation analysis presented in the Table 1.

Reviewer 2 Report

The manuscript is interesting and may be published after minor revision. Some parts need to be corrected/improved.

General remark - could you provide any data showing changes in volcanic activity within the investigated period? 

31-36: This is basic palaeoecological knowledge, not related to the topic of the manuscript so I suggest deleting it. More sense would have writing about present field experiments conducted on mires like eg. those described in a Climpeat project - eg. how artificial manipulation in water table depth on peatland affects vegetation change (eg. http://dx.doi.org/10.19189/MaP.2016.OMB.244)

65 - "we" - there is only one author of this manuscript unless the author thinks about members of the laboratory mentioned in the acknowledgments

2.1. I suggest adding altitude to each site description

Fig 1. is not clear and should be redrawn. In fact, there are three maps in the figure but it is not easy to understand where is the boundary of each map. "The second" map shows probably some administrative borders but at first glimpse, it looks like a coast of an island (maybe you may fix it by adding colors or labels). As these are volcanic mires I assume that you may get a nice visual effect if you use a digital elevation model as a background to the most detailed map (SRTM resolution should be sufficient).

124-125 "Data from August 2008 to May 2010 were not presented in the TDW transect, and data from August 2007 to July 2010 were not presented in the BGT transects 1 and 2". - Why?

What software was used to draw figures 3-8?

213-214: I do not understand it. Decreased in general, but increased in a particular period and part? This requires clarification

Was 2010 somehow different from the other years, that sulfate ion concentration increased in TDW transect? Any significant change in volcanic activity occurred then?

Why TOC had the highest values on each site in 2012?

Table 7 - add information about the meaning of red and blue color

Author Response

Reply to the Reviewer #2

Thank you very much for your constructive comments on my manuscript: water-2081383. I have made revision on my manuscript according to your suggestion.

The manuscript is interesting and may be published after minor revision. Some parts need to be corrected/improved.

Answer: Thank you very much for evaluating my paper.

General remark - could you provide any data showing changes in volcanic activity within the investigated period?

Answer: There was no record of eruption of volcano in the investigated area during the investigated period. Japan Meteorological Agency announced the prediction of eruption on 1st December in 2007; however, the probability of eruption is the lowest level and there was no discernible change (earthquakes) at that moment. So, I decided that the volcanic activity during the investigated period was fairly stable. I added the description in ll. 416–425; 'Monthly average atmospheric temperature and monthly precipitation in Kusu Mete-orological Observatory located near the Tadewara mire and the Bougatsuru mire (33.292°N, 131.155°E, 331 m a.s.l.; https://www.data.jma.go.jp/obd/stats/etrn/index.php?prec_no=83&block_no=0932&year=2022&month=&day=&view=) showed no significant secular change from 2006 to 2016. Eruption or discernible increase of volcanic earthquake and volcanic smoke height from fumarole were never reported in Kujusan (nearest observation station of volcano from the investigation sites) from 2006 to 2016  (https://www.data.jma.go.jp/vois/data/fukuoka/502_Kujusan/502_index.html). Therefore, direct effects of climate change and volcanic activity on vegetation in the investigated mires are regarded to be limited.'

31-36: This is basic palaeoecological knowledge, not related to the topic of the manuscript so I suggest deleting it. More sense would have writing about present field experiments conducted on mires like e.g. those described in a Climpeat project - eg. how artificial manipulation in water table depth on peatland affects vegetation change (eg.http://dx.doi.org/10.19189/MaP.2016.OMB.244)

Answer: I deleted the paragraph in ll. 31-36. I also deleted the literature No. 5, 6, and 7, and renumbered the citation list. 'Climpeat project' is important; however, artificial manipulation of water table is a little bit different topic from volcanic activity. Therefore, I only deleted the paragraph.

65 - "we" - there is only one author of this manuscript unless the author thinks about members of the laboratory mentioned in the acknowledgments

Answer: I replaced 'we' by 'I' in l. 64. And I checked the whole document.

2.1. I suggest adding altitude to each site description

Answer: I have added the altitude of the center of the investigated sites; 1017 m a.s.l. for the Tadewara mire (l. 72) and 1240 m a.s.l. for the Bougatsuru mire (l. 82).

Fig 1. is not clear and should be redrawn. In fact, there are three maps in the figure but it is not easy to understand where is the boundary of each map. "The second" map shows probably some administrative borders but at first glimpse, it looks like a coast of an island (maybe you may fix it by adding colors or labels). As these are volcanic mires I assume that you may get a nice visual effect if you use a digital elevation model as a background to the most detailed map (SRTM resolution should be sufficient).

Answer: Thank you very much for your useful suggestion. I think that the digital elevation model will provide excellent map in the investigated area. However, I am sorry but I do not have ability to prepare the map. Therefore, I only redrawn the map deleting the first small map of Japan Islands. Coordination in the right side map in the revised Figure 1 will give the rough location of the investigated site. Information on the exact position of the investigated sites appear in the text (ll. 97-106).

124-125 "Data from August 2008 to May 2010 were not presented in the TDW transect, and data from August 2007 to July 2010 were not presented in the BGT transects 1 and 2". -Why?

Answer: That is because of the problem on research budget. But I only mention that the data is missing during this period. The main analysis of my paper is the comparison between 2006 and 2011 as well as comparison between 2011 and 2016. And so, the missing period is not so much affecting the statistical analysis.

What software was used to draw figures 3-8?

Answer: I used 'R version 4.2.1'. I changed the sentence in ll. 188-189 as 'Missing data were interpolated, and contour figures were drawn by R version 4.2.1. [13]', and sentence in l. 237 as ' Missing data were interpolated, and contour figures were drawn by R version 4.2.1. [13]'. Further, I added the citation as literature No. [13].

13          R Core Team (2022). R: A language and environment for statistical computing. R Foundation for Statistical Computing, Vienna, Austria. URL https://www.R-project.org/.

213-214: I do not understand it. Decreased in general, but increased in a particular period and part? This requires clarification

Answer: I am sorry but I have made a mistake. I have changed the sentence in ll. 213–214 as 'The coverage of the two Sphagnum spp. decreased from 2006 to 2011 and increased from 2012 to 2016, particularly at the upper part (10–20 m sites) of the BGT transect 2 (Figure 5).

Was 2010 somehow different from the other years, that sulfate ion concentration increased in TDW transect? Any significant change in volcanic activity occurred then?

Why TOC had the highest values on each site in 2012?

Answer: As I mentioned in the answer to your General remark, there was no record of eruption of volcano in the investigated area during the investigated period. And I decided that the volcanic activity during the investigated period was fairly stable. Increase in sulfate ion concentration would be due to the increase of supply of sulfate from spring water. High TOC value in 2012 would be due to the rapid decrease of water table depth and the consequence increase of aeration and decomposition rate of organic materials. But both the discussion on sulfate and TOC change are lacking evidence. Therefore, I only mention the explanation of data in ll. 433–436 (The Ca2+ concentration decreased from 2010, corresponding to the temporal SO42- increase and a pH decrease in 2010. Inundated water at the 0–100 m TDW transect sites supplies sulfuric acid, and Ca2+ would be removed through CaSO4 precipitation.), and ll. 441–442 (WTD was the lowest during the 2012–2014 period, corresponding to the high TOC owing to the in-creased decomposition of organic materials in the soil.).

Table 7 - add information about the meaning of red and blue color

Answer: I have added the sentence 'Red colored cells mean significant positive correlations and blue colored cells mean significant negative correlations.' in ll. 409–410 in Table 7.
